# A Cooperation Index for Model Pruning

## Abstract

In complex models, tools for measuring parameter importance identify its core functional element and improve both generalizability and interpretability by pruning redundant ones. Effective pruning relies on these tools, which serve as decision making criteria. The SHAP Value (SV) has recently been considered such a criterion, interpreted as measuring the average marginal contribution across all possible paths of parameter accumulation. However, we find that this averaging process of SV systematically overweights redundant parameters. Instead, we propose that measuring the speed of decay of the marginal contribution can serve as a more effective decision-making criterion. Specifically, we quantify the number of cooperative contribution for each parameter and show that this criterion is more effective for parameter pruning in backward elimination, leading to a more optimal set of remaining parameters.

## 1 Introduction

Neuroplasticity is one of the fundamental properties of neural networks Hassibi et al. (1993); Lecun et al. (1989). As human brain adapts by reinforcing useful connections and compensates for a deficiencies or losses through neuroplasticity, a similar process can benefit artificial neural networks by pruning unnecessary parameters and optimizing the remaining ones Han et al. (2015); Li et al. (2017); Luo et al. (2017); Molchanov et al. (2017); Yeh et al. (2019); Ghorbani & Zou (2020). The interactions among parameters, however, are often complex and non-additive, making it challenging to assess the individual contribution of each parameter within a complex model. Some parameters are cooperative with others, consistently enhancing the model's predictive capabilities, while redundant ones are easily replaceable by other parameters.

One widely adopted and influential tool for measuring feature contribution is the SHAP value (SV) defined as Lundberg & Lee (2017); Zaeri-Amirani et al. (2018); Cohen et al. (2007); Tripathi et al. (2021); Lecun et al. (1989); Marcílio & Eler (2020):

$$\phi_i(f) = \frac{1}{n} \sum_{S \subseteq \mathcal{P} \setminus \{i\}} \left( \begin{array}{c} n-1 \\ |S| \end{array} \right)^{-1} (f(S \cup \{i\}) - f(S)). \tag{1}$$

Given a set of features, $\mathcal{P}$, of size $n$, SV $\phi_i(f)$ represents the weighted average contribution of feature $i$ by considering all possible cases in which feature $i$ is additionally applied to the feature subset $S \subseteq \mathcal{P} \setminus \{i\}$. Here, the term $(f(S \cup \{i\}) - f(S))$ in Eq 1 is dubbed as marginal contribution of feature $i$. Note that this term is not fixed; instead, it changes depending on the feature subset $S$. The SV is predominantly used as sensitive, high-risk decision-making agents across broad real-world problems due to its game-theoretically principled nature Lundberg & Lee (2017); Zaeri-Amirani et al. (2018); Cohen et al. (2007); Tripathi et al. (2021); Marcílio & Eler (2020). However, our main observations show that SV's core concept—averaging—overweights redundant parameters and can lead to counter-intuitive decisions.

For the two graphs illustrated in Figure 1, when the marginal contribution of a parameter is represented across various combinations of other parameters, and the contribution values are sorted in descending order from the left to right. The average contributions (SVs mentioned in Eq. 1) of both are same, but the **decay patterns** of the marginal contribution differ significantly. In the left graph Fig. 1a, when the parameter is additionally applied, the decay pattern of the marginal contribution depicts high contributions but only in limited combinations of other parameters, showing the speed of decay is fast. In contrast, the right graph Fig. 1b illustrates that although the parameter's marginal contributions

are not exceptionally high, they consistently contribute across numerous combinations of other parameters, showing the speed of decay is slow. Pruning the parameter on the left can be easily offset by others, whereas pruning the one on the right may lead to a significant loss in overall performance. The SV, by averaging contributions, fails to distinguish between these two fundamentally different roles.

These observations point to a critical consideration in model pruning: not all parameters with a similar average contribution play equivalent roles within the networks. Instead, our motivation is that parameters consistently contributing in conjunction with various combinations of other parameters are unlikely to be replaced by other parameters and must therefore be retained within the networks. Recognizing these difference of the roles is crucial, yet effective methods for leveraging this information remain largely underdeveloped. To address this gap, we propose a simple index that quantifies the speed of decay of the marginal contribution when trained alongside different combinations of other parameters.

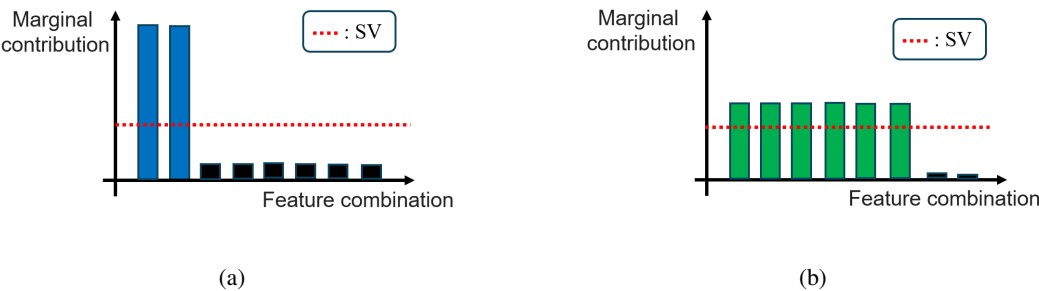

(a)                                                 (b)

Figure 1: A conceptual illustration of decay patterns of the marginal contribution for two different parameters. Each bar represents a marginal contribution corresponding to different parameter combinations. With black bars indicating insignificant contributions.

Recent advances in neural networks have been powered by the extensive use of computational resources to expand model sizes. However, increasing attention is being given on evaluating models based on their structural efficiency while ensuring that performance remains uncompromised Belkin et al. (2019); Han et al. (2015); Hassibi et al. (1993); Lecun et al. (1989); Jacot et al. (2020); Li et al. (2017); Luo et al. (2017); Neyshabur et al. (2015); Soudry et al. (2024); Frankle & Carbin (2019); Ramanujan et al. (2020); Zhou et al. (2020). These studies have shown that many parameters can be removed with minimal or even no impact on performance. This functional redundancy among parameters has prompted extensive interest in model compression. In response, methods for scoring parameter importance have been developed to identify key functional parameters within models Lecun et al. (1989); Han et al. (2015); Yeh et al. (2019); Molchanov et al. (2017); Bau et al. (2017); Ghorbani & Zou (2020). Several recent studies have focused on the Shapley value (SV) due to its well-defined game-theory axioms for selecting important features or parameters Lundberg & Lee (2017); Zaeri-Amirani et al. (2018); Cohen et al. (2007); Tripathi et al. (2021); Lecun et al. (1989); Marcílio & Eler (2020). However, they also raise concerns about the over-reliance on SV applications, pointing out a critical flaw: their inability to account for redundant features effectively Fryer et al. (2021); Ma & Tourani (2020).

The remainder of the paper is organized as follows. Section 2 delves into the SV and its interpretation, explaining how it and related methods use marginal contributions to measure feature importance. Section 3 introduces a simple criterion for selecting parameters for model pruning. Following this, we describe the implementation of the proposed criterion, showcasing experimental results in Section 4, and conclude with a summary in Section 5.

## 2   MARGINAL CONTRIBUTION AND PARAMETER IMPORTANCE

The contribution of a parameter is inherently tied to its interaction with other parameters. Accordingly, we consider the accumulation of contributions by arranging the parameters in a set of parameter $\mathcal{P}$ into all possible permutations. Along the permutation sequence, we sequentially add parameters one

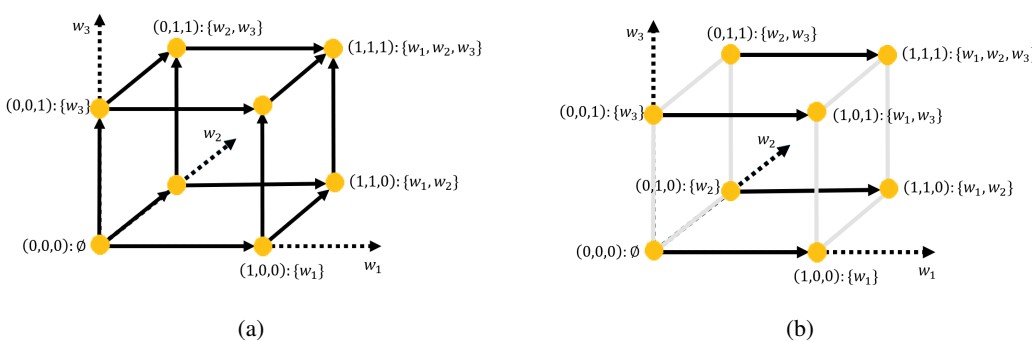

Figure 2: (a) A three-dimensional cube representing different edge paths from $(0,0,0)$ to $(1,1,1)$ corresponding to a vector of three parameters $(w_1, w_2, w_3)$. (b) Visualization of all paths in which $w_1$ can contribute in conjunction with the parameters encountered before $w_1$.

at a time. At each step of the sequence, the model consisting of all parameters added up to that step is jointly optimized on the data.

Let $\Pi(\mathcal{P})$ denote the set of all possible permutations of the parameter indices $i \in \mathcal{P}$ and define $S_\pi^i$ as the subset of parameters that appeared before $i$ in the ordering $\pi \in \Pi(\mathcal{P})$. We then define $f^*(S_\pi^i)$ as the objective function—such as the log likelihood or the negative loss function—optimized with respect to the data over the parameters in $S_\pi^i$.

**Marginal Contribution.** Given the permutation $\pi \in \Pi(\mathcal{P})$, the marginal contribution of parameter $i \in \mathcal{P}$ is defined as:

$$\Delta_{\pi,i} = f^*(S_\pi^i \cup \{i\}) - f^*(S_\pi^i), \tag{2}$$

similar to Shapley (1953). The marginal contribution $\Delta_{\pi,i}$ quantifies the performance gain achieved by adding parameter $i$ along that permutation $\pi$. The $\Delta_{\pi,i}$ is nonnegative because, with the addition of parameter $i$, the worst-case performance of $f^*(S_\pi^i \cup \{i\})$ is $f^*(S_\pi^i)$, which occurs when parameter $i$ is not utilized.

**Geometrical Interpretation.** Geometrically, for a given parameter permutation, the process of sequential parameter addition can be interpreted as a shortest path, traversing various intermediate vertices along the edges of an $n$-dimensional hypercube, as shown in Fig. 2a from the origin $\mathbf{0}_n = (0, \ldots, 0)^\top$ to $\mathbb{1}_n = (1, \ldots, 1)^\top$ Candogan et al. (2011); Stern & Tettenhorst (2019). Each vertex represent the parameter subsets $S$, each with a performance value $f^*(S)$ trained with the parameter in the subset $S$. The marginal contribution $\Delta_{\pi,i}$ (Eq. 2) is then precisely the change in $f^*$ along an edge of this hypercube, corresponding to the addition of parameter $i$ (Fig. 2b). Note that

$$\sum_{i \in \mathcal{P}} \Delta_{\pi_1,i} = \sum_{i \in \mathcal{P}} \Delta_{\pi_2,i} = f^*(\mathcal{P}), \tag{3}$$

for any $\pi_1, \pi_2 \in \Pi(\mathcal{P})$ because the sum of marginal contributions of every parameters along any permutation path $\pi \in \Pi(\mathcal{P})$ equals the value of $f^*(\mathcal{P})$ optimized with all parameters, corresponding to the vertex $\mathbb{1}_n$. The main point is that different permutation paths may give different marginal contributions for the same parameter, which gives a non-uniform decay pattern as shown in Fig. 1.

**SHAP Value (SV) Covert et al. (2020); Lundberg & Lee (2017); Shapley (1953):** The SV in Eq. (1) can be rewritten as a simple average of the marginal contributions across all possible permutation paths on the $n$-dimensional hypercube:

$$\phi_i = \frac{1}{n!} \sum_{\pi \in \Pi(\mathcal{P})} \Delta_{\pi,i}, \tag{4}$$

where the $f$ in Eq. (1) is now replaced by the optimized objective $f^*$. A simple example, pruning parameter choice under the vertex values detailed in Appendix A, illustrates that SV can result in clearly wrong decisions.

## 3 COOPERATION INDEX

### 3.1 DEFINITION

We now formally define the Cooperation Index (CI). The definition is based on classifying the characteristic of a parameter's marginal contribution within each permutation path $\pi \in \Pi(\mathcal{P})$.

- **Cooperative Path:** Parameter $i \in \mathcal{P}$ is said to be cooperative if its marginal contribution satisfies $\Delta_{\pi,i} > \phi_i$. In this case, its marginal contribution is said to be cooperative contribution. If the number of cooperative path is high, parameter $i$ consistently contributes more than its expected value, resulting in the speed of decay of marginal contribution being slow.

- **Replaceable Path:** Parameter $j \in \mathcal{P}$ is said to be replaceable if marginal contribution satisfies $\Delta_{\pi,j} < \phi_j$. If the number of replaceable paths is high, the SV assigned to this parameter is easily achieved by other parameters, indicating that this parameter is replaceable by other parameters. In this case, the speed of decay of marginal contribution is fast.

Based on the number of cooperative paths on the $n$-dimensional hypercube, we define the Cooperation Index (CI) for parameter $i$ as follows:

$$\text{CI}(i) = \frac{|\{\pi \in \Pi(\mathcal{P}) : \Delta_{\pi,i} > \phi_i\}|}{|\Pi(\mathcal{P})|} = \frac{1}{n!} \sum_{\pi \in \Pi(\mathcal{P})} \mathbf{1}(\Delta_{\pi_j,i} > \phi_i). \tag{5}$$

Here, $\Pi(\mathcal{P})$ denotes the set of all permutation paths over the parameter set $\mathcal{P}$. If the total number of parameters is $n$, the denominator $(|\Pi(\mathcal{P})|)$ is $n!$. Note that SV and CI are used for parameter ranking purposes, and both methods remove low-ranking parameters. The computational cost for calculating both is the same.

### 3.2 PERFORMANCE RETENTION UNDER PRUNING

Since the SV is defined as the average of marginal contributions (Eq. 4) and the sum of all SVs corresponds to the performance of the model when all parameters are used, we consider two model performance measures, with and without parameter $k$:

$$J_{\text{tot}}(\mathcal{P}) = \sum_{i \in \mathcal{P}} \phi_i, \quad J_{\text{tot}}(\mathcal{P} \backslash \{k\}) = \sum_{i \in \mathcal{P} \backslash \{k\}} \phi_i^{\sim k}. \tag{6}$$

Under the following replaceability assumption where parameter $k$ does not contribute in the presence of any $l \in \mathcal{P} \backslash \{k\}$ for any subset $S \subseteq \mathcal{P} \backslash \{l, k\}$,

$$f(\{k\} \cup S \cup \{l\}) - f(S \cup \{l\}) = 0, \tag{7}$$

we can derive that the SV of parameter $l$ increases after pruning $k$, i.e., $\phi_l^{\sim k} \geq \phi_l$ for all such $l$.[1]

This inequality implies that eliminating parameter $k$ is expected to reduce $J_{\text{tot}}(\mathcal{P})$ by the amount of $\phi_k$, but this loss is compensated by an increase in the SV of the remaining parameters, and $J_{\text{tot}}(\mathcal{P} \backslash \{k\}) > J_{\text{tot}}(\mathcal{P}) - \phi_k$ by pruning *replaceable* parameters. In this way, the contribution originally attributed to $k$ does not vanish after pruning—it is redistributed among the others when the replacement property is considered. This property shows that SV wrongly determines that the replaceable parameter $k$ is as important as $\phi_k$, which can be compensated by others. Instead, the CI assigns a low importance score to parameter $k$ by measuring the speed of decay of marginal contribution due to the large number of replaceable paths.

In terms of distribution of the marginal contribution, pruning a parameter removes all permutations that include the eliminated parameter as illustrated in Figure 3. If the average of the marginal contribution over the remaining permutations exceeds the marginal contributions lost due to pruning, then the overall impact of pruning can be minimal.

---

[1]See Appendix B for details.

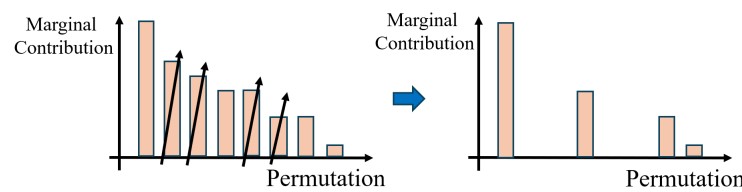

Figure 3: Illustration of the process of eliminating marginal contributions associated with the removed parameters after pruning. Each bar represents the marginal contribution obtained from a single permutation.

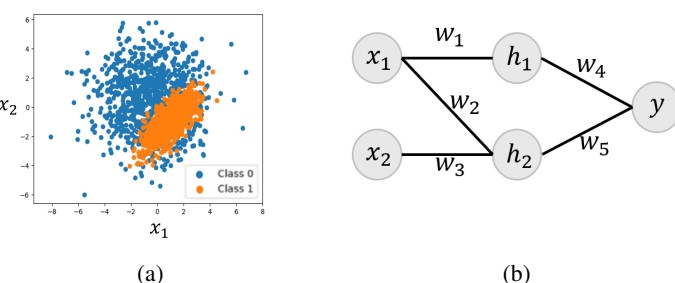

(a)

(b)

Table 1: Importance scores for the initial stage

| Params | SV | CI |
|--------|--------|------|
| $w_1$ | 0.0790 | 0.25 |
| $w_2$ | 0.0836 | 0.25 |
| $\mathbf{w_3}$ | **0.0350** | **0.38** |
| $w_4$ | 0.0790 | 0.25 |
| $w_5$ | 0.1134 | 0.25 |

Figure 4: (a) Training data: two-class 2-D gaussian. (b) A toy neural network consisting of five weight parameters $w_1, \ldots, w_5$ and a single hidden layer with hidden units $h_1$ and $h_2$.

### 3.3 AN ILLUSTRATIVE EXAMPLE WITH SYNTHETIC DATA

Here, we present a simple toy example to illustrate the central concept of the CI. The neural network in Figure 4b has five parameters, $w_1, \ldots, w_5$, which yields $2^5 = 32$ parameter subsets and $5! = 120$ permutations. In each permutation path, the subset of learnable parameters is trained on the training data in Figure 4a to predict the labels, while the parameters excluded from the subset are fixed at zero and retained in the network. After training, the model's negative cross-entropy is evaluated on the training data as a performance measure. Initially, the SV and CI scores for all five parameters are presented in Table 1. We note that parameter $w_3$ exhibits the *largest* CI, while the others exhibit the *lowest*. In the initial pruning, $w_3$ is never pruned by the CI criterion, whereas SV-based criterion select $w_3$ for pruning. Once parameter $w_3$ is removed, the classification information from $x_2$ is lost. These cooperative behavior of $w_3$ appears as a high CI score for the given training data. According to Figure 5 and 6, the order of parameter pruning by CI is $w_4 \to w_1 \to w_3 \to w_2 \to w_5$. Figure 5 shows that after $w_4$ is removed in the first pruning stage, $w_1$ loses all cooperative parameters and becomes the eliminated parameter in the next pruning stage, while $w_2$ and $w_5$ strengthen their cooperation, leading to an increase in both of their CI scores. Figure 6 illustrates the dynamics of parameter's scores in the SV-CI space as each parameter is removed.

### 3.4 TWO-LEVEL APPROXIMATION SCHEME FOR THE CI ESTIMATION

As in the conventional calculation of SV and CI, we approximate the computation of marginal contributions over all subsets by sampling from $n!$ permutations to avoid an exponential complexity of $O(2^n)$ Lundberg & Lee (2017); Catav et al. (2021). In addition to permutation sampling, it would require evaluating the model performance $f^*(S)$ for every subset in every sampled permutation. To make this feasible, we apply regression on the $f^*$-hypercube to approximate the marginal contributions and introduce a two-level approximation scheme as illustrated in Figure 7.

**Level 1: Permutation sampling.** For the random sampling of the permutations $\Pi_{\text{samples}} \subseteq \Pi(\mathcal{P})$ Lundberg & Lee (2017); Castro et al. (2009), we estimate the marginal contribution for all

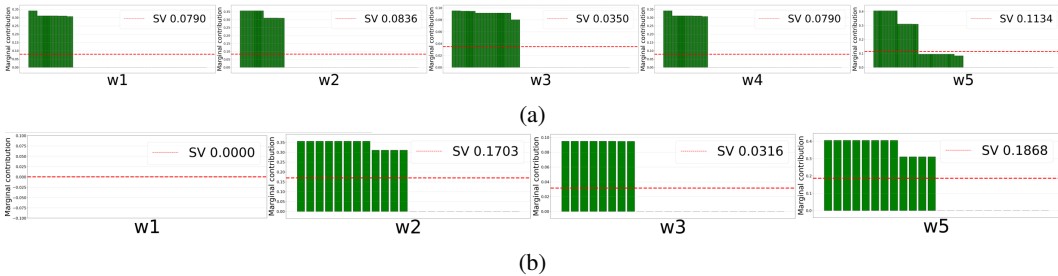

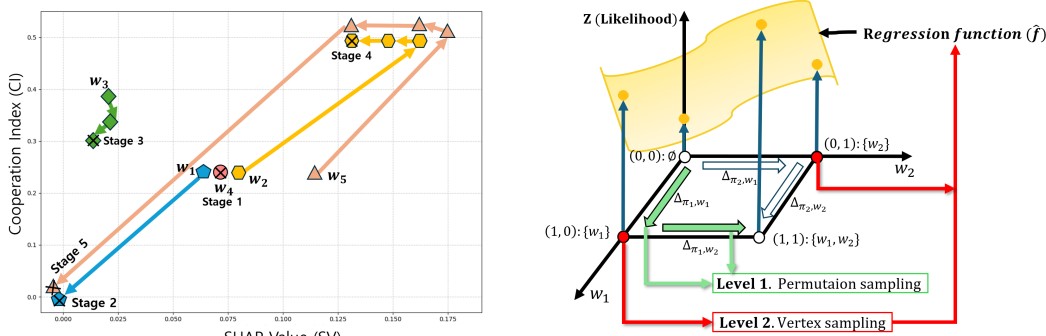

Figure 5: Marginal contributions across all permutations and SV for the five parameters at the initial and second pruning stages. (a) Initial pruning stage. (b) Second pruning stage after $w_4$ has been removed.

Figure 6: Trajectories of the SV and CI scores of the each parameters.

Figure 7: Illustration of the two-level approximation scheme for the CI estimation.

parameter indexes $i \in \mathcal{P}$,

$$\widehat{\Delta}\pi, i = \widehat{f}(S_\pi^i \cup i) - \widehat{f}(S_\pi^i), \tag{8}$$

for any $\pi \in \Pi_{\text{samples}}$. Here, $\widehat{f}(S)$ is regression function over the $n$-dimensional hypercube vertices to approximate $f^*(S)$. The complexity of level 1 is $O(N \cdot S)$, where M is the number of permutations and S is the number of parameters.

**Level 2: Vertex value prediction.** Directly calculating $f^*(S)$ for every required subset $S$ is another computational bottleneck. The purpose of Level 2 is to create a regression function $\widehat{f}(S)$ to overcome this challenge. We sample $m$ number of vertices of the $f^*$-hypercube and train the corresponding parameter subset $S$ to obtain $f^*(S)$. Those $f^*(S)$ over the hypercube vertices are trained with a regression model $\widehat{f}(S)$ as illustrated in Figure 7. The complexity of level 2 is $O(m \cdot T_{eval})$, where m is the number of sampled vertices (subsets) and $T_{eval}$ is the time to evaluate one subset.

Using the estimated marginal contribution $\widehat{\Delta}_{\pi,i}$, SV and CI are calculated as following:

$$\widehat{\phi}_i = \frac{\sum_{\pi \in \Pi_{\text{sample}}} \widehat{\Delta}_{\pi,i}}{|\Pi_{\text{sample}}|}, \qquad \widehat{\text{CI}}(i) = \frac{\sum_{\pi \in \Pi_{\text{sample}}} \mathbf{1}(\hat{\Delta}_{\pi,i} > \hat{\phi}_i)}{|\Pi_{\text{sample}}|}. \tag{9}$$

SV is the average of the marginal contributions over permutation samples while the CI measures how fast the pattern of marginal contributions decay. The following is detailed algorithm for two-level approximation scheme.

## 4 EXPERIMENTS

We assess the effectiveness of the Cooperation Index (CI) for model pruning tasks and benchmark it against other baseline importance scoring methods. We conduct real-world experiments on large-scale models and diverse datasets to demonstrate the applicability and scalability of our approach.

---

**Algorithm 1** CI Calculation via Two-Level Approximation

---

**Require:**
 1: $N$: The trainable neural network.
 2: $P$: The set of all prunable parameters, total number $|P|$.
 3: $N_v$: Number of vertex (parameter subset) samples.
 4: $N_p$: Number of permutation samples.
 5: $\hat{f}$: Trained regression model (from Level 2).
**Ensure:**
 6: $CI_{scores}$: Dictionary of CI scores for each parameter.

    *Part 1: Calculate all marginal contributions*
 7: $C_{\text{full}} \leftarrow$ Initialize empty list for each parameter $p \in P$.
 8: **for** $j \leftarrow 1$ to $N_p$ **do**
 9:    $\pi \leftarrow$ Randomly permute the set $P$ (Level 1).
10:    $S_{\text{prev}} \leftarrow \emptyset$
11:    **for** $k \leftarrow 1$ to $|P|$ **do**
12:        $p_k \leftarrow$ The $k$-th parameter in permutation $\pi$.
13:        $S_{\text{curr}} \leftarrow S_{\text{prev}} \cup \{p_k\}$
14:        $\Delta \leftarrow \hat{f}(S_{\text{curr}}) - \hat{f}(S_{\text{prev}})$
15:        Add $\Delta$ to the list $C_{\text{full}}[p_k]$
16:        $S_{\text{prev}} \leftarrow S_{\text{curr}}$
17:    **end for**
18: **end for**

    *Part 2: Compute CI scores from contributions*
19: $CI_{scores} \leftarrow$ Initialize empty dictionary.
20: **for** each parameter $p$ in $P$ **do**
21:    $SV_p \leftarrow \text{Mean}(C_{\text{full}}[p])$
22:    $count_{\text{cooperative}} \leftarrow 0$
23:    **for** each contribution $\Delta$ in $C_{\text{full}}[p]$ **do**
24:        **if** $\Delta > SV_p$ **then**
25:            $count_{\text{cooperative}} \leftarrow count_{\text{cooperative}} + 1$
26:        **end if**
27:    **end for**
28:    $CI_{scores}[p] \leftarrow count_{\text{cooperative}}/|C_{\text{full}}[p]|$
29: **end for**

30: **return** $CI_{scores}$

---

We use VGG-16 Simonyan & Zisserman (2014) and ResNet-18 He et al. (2016) architectures for experiments. The experiments are designed to confirm generalization improvement stemming from the initial, delicate filter removal. We intentionally overfit the models by reducing training data and perform model pruning at the individual filter level in the all experiments. The core purpose of our experiment is to confirm generalization performance gains and performance preserving at the start of the pruning performed by each methods, rather than high-ratio pruning. Here, the pruning ratio indicates the percentage of the removed filters. The experiments are conducted five realizations with different random seeds, and we report the best performing result in terms of accuracy. When we evaluate the pruned model, the weights of removed filters are fixed as zero and retained in the model.

**Baselines.** We compare our CI method against other baseline criteria. The SV Lundberg & Lee (2017) and CI use a common set of samples of the marginal contribution estimations. In addition, we present the results of widely used model pruning methods, such as those based on parameter magnitudes Han et al. (2015); Frankle & Carbin (2019), Leave-One-Covariate-Out Lei et al. (2017) and network slimming Li et al. (2017). Note that the Marginal Contribution feature Importance (MCI) Catav et al. (2021) which is a criterion that uses a common set of samples of the marginal

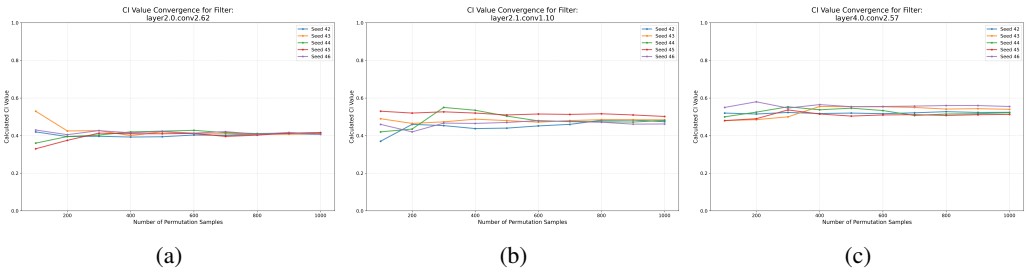

Figure 8: Convergence of CI.

contribution estimations are not considered a baseline in real-world experiments because they show poor performance in the synthetic experiment.[2]

**Datasets.** To intentionally overfit the model to the training datasets, we used MNIST LeCun et al. (2010), CIFAR-10 Krizhevsky et al. (2009), CIFAR-100 Krizhevsky et al. (2009) and Tiny-ImagNet Deng et al. (2009) datasets. For MNIST experiments, Both models were trained on the dataset scaled down to 1/100 of the original training dataset. For CIFAR-10 experiments, Both models were trained on the dataset scaled down to 1/2 of the original training dataset. For CIFAR-100 and Tiny-ImageNet experiments, Both models were trained on the dataset scaled down to 4/5 of the original training dataset.

**Pruning Protocols and Evaluation.** We follow the two-level approximation scheme described in Section 3.4.

- **Level 2 (Vertex value prediction):** We employ a fully connected Multi-Layer Perceptron (MLP) as the regression model. Specifically, the architecture consists of two hidden layers, each with 4096 neurons, using ReLU activation functions. The hyperparameter includes the learning rate (1e-4), and training epochs (100). To train the regression model $\widehat{f}(S)$, we first generate a dataset of performance values $f^*(S)$. This is done by sampling parameter subsets $S$, where each filter is included in a subset with a probability of $p = 0.5$. For each subset, the filters in the subset are kept active while the others are fixed to zero, and the resulting sub-networks is trained on the given training datasets to obtain its performance $f^*(S)$—such as the log likelihood or the negative loss function.

- **Level 1 (Permutation Sampling):** Using the regression model obtained from level 2, we estimate marginal contributions by sampling permutations of whole parameters. We empirically verified that the variance of the CI score can be stabilized by sampling enough permutations, providing a reliable ranking of parameter importance.

- **Pruning and Evaluation:** In our implementation, pruning is performed in a single step for each target pruning ratio. To achieve a desired pruning ratio, we first rank all filters by their CI scores in ascending order. We then select the batch of filters with the lowest CI scores corresponding to target ratio and remove them all at once. Thus, model pruning is implemented by fixing all weights in the lowest-ranked filter batch to zero, and the pruned model is evaluated on the test datasets to obtain the test accuracy. Additionally, the experiments took approximately 4 GPU-hours on a single NVIDIA A6000 GPU and required about 2GB of GPU memory.

**Stability of CI.** We observed how the CI value differ by the number of permutations sampled in level 1. As shown in Fig 8, we sampled permutations between 100 and 1000 times from different random seeds and calculated CI values for three randomly selected filters of the ResNet-18 model. When sampling 1000 permutations, the CI almost converged to a single value empirically. Therefore, we empirically sampled 1000 permutations for calculating CI values in all experiments. Additionally, we derived a theorem for the convergence of the CI and showed that the convergence of CI is independent of the model size. Therefore, the CI converges even if the model scales up.[3]

---

[2]See Appendix C for details.
[3]See Appendix D for details.

Table 2: Comparison of Pruning Methods.

| Model, Datasets | Method | Original Accuracy | Accuracy at Pruning Ratio |
|---|---|---|---|
| VGG-16, MNIST | SV Lundberg & Lee (2017) | 0.882 | 0.877 at 3% |
| | L1-Norm Frankle & Carbin (2019) | 0.882 | 0.098 at 3% |
| | LOCO Lei et al. (2017) | 0.882 | 0.729 at 3% |
| | Slimming Liu et al. (2017) | 0.882 | 0.626 at 3% |
| | CI (Ours) | 0.882 | **0.925** at 3% |
| VGG-16, CIFAR-10 | SV | 0.846 | 0.812 at 3% |
| | L1-Norm | 0.846 | 0.010 at 3% |
| | LOCO | 0.846 | 0.820 at 3% |
| | Slimming | 0.846 | 0.515 at 3% |
| | CI (Ours) | 0.846 | **0.836** at 3% |
| ResNet-18, MNIST | SV | 0.876 | 0.771 at 3% |
| | L1-Norm | 0.876 | 0.098 at 3% |
| | LOCO | 0.876 | 0.767 at 3% |
| | Slimming | 0.876 | 0.707 at 3% |
| | CI (Ours) | 0.876 | **0.830** at 3% |
| ResNet-18, CIFAR-10 | SV | 0.826 | 0.789 at 3% |
| | L1-Norm | 0.826 | 0.010 at 3% |
| | LOCO | 0.826 | 0.738 at 3% |
| | Slimming | 0.826 | **0.819** at 3% |
| | CI (Ours) | 0.826 | 0.807 at 3% |
| ResNet-18, CIFAR-100 | SV | 0.752 | 0.662 at 2% |
| | LOCO | 0.752 | 0.692 at 2% |
| | CI (Ours) | 0.752 | **0.710** at 2% |
| ResNet-18, Tiny-ImageNet | SV | 0.613 | 0.543 at 1% |
| | LOCO | 0.613 | **0.553** at 1% |
| | CI (Ours) | 0.613 | 0.543 at 1% |

**Results.** The experimental results in Table 2 demonstrate that CI consistently preserves the core functional elements of the model in comparison with baseline methods. The experiment of the VGG-16 model on the MNIST dataset reveals an interesting result that removing unnecessary parameters (filters) can improve the generalization performance. In particular, for relatively simple datasets MNIST and CIFAR10, CI demonstrates superior results compared to other methods by identifying unnecessary filters that overlap in function with other filters. Therefore, the CI demonstrates its effectiveness as a decision-making agent for model pruning. The L1-Norm and Slimming methods among the baseline approaches perform poorly on relatively simple datasets, MNIST and CIFAR-10, and are therefore excluded from consideration in experiments on more expanded datasets. The results on the Tiny-ImageNet dataset demonstrate pathological case that CI can underperform when each filter plays a distinct role with little redundancy, however, it achieves at least the same performance as SV.

## 5 CONCLUSION

This study introduces a novel and simple criterion for measuring parameter importance. The Cooperation Index (CI) quantifies the speed of decay of the marginal contribution and addresses the limitation of SHAP value. This approach is effective for model pruning, revealing model's core functional elements and improving the generalizability of the model. A key challenge moving forward lies in adapting Cooperation Index to larger models and using more diverse datasets across a range of applications.

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
