# A APPENDIX

**An Example of SV-Based Pruning Failure.** Three parameters, denoted by $w_1, w_2, w_3$, produce a total gain of 10 when all are used. Let they have an $f^*$-hypercube with the following vertex values:

$$f^*(S) = \begin{cases} 10, & \text{if } S = \{w_1, w_2, w_3\} \\ 10, & \text{if } S = \{w_1, w_2\}, \\ 10, & \text{if } S = \{w_1, w_3\}, \\ 7, & \text{if } S = \{w_2, w_3\}, \\ 7, & \text{if } S = \{w_2\}, \\ 7, & \text{if } S = \{w_3\}, \\ 0, & \text{if } S = \{w_1\}, \\ 0, & \text{if } S = \emptyset. \end{cases} \tag{1}$$

The corresponding $f^*$ values represent the incremental gains associated with transitions along vertex paths. The SVs assigned to each parameter are $(w_1, w_2, w_3) = (2, 4, 4)$, indicating that $w_1$ should be pruned first. On the other hand, CI yields $(w_1, w_2, w_3) = \left(\frac{2}{3}, \frac{1}{2}, \frac{1}{2}\right)$ reflecting the redundancy among parameters. According to the CI, $w_2$ or $w_3$ should be pruned, as they are replaceable by each other, preserving the total gain after pruning.

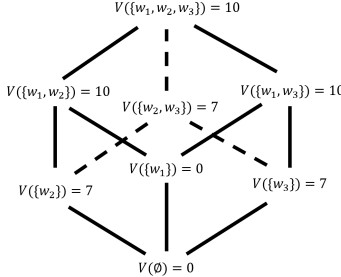

(a) geometric view of all possible path

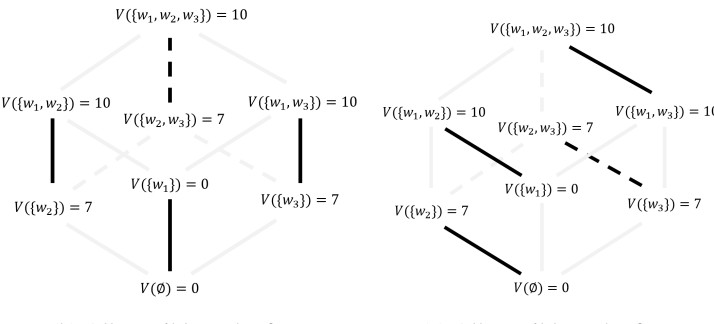

(b) All possible path of $w_1$        (c) All possible path of $w_2$

Figure 1: Visualization of the $f^*$-hypercube in Appendix A.

## B  APPENDIX B

**The Impact of Parameter Removal on SHAP Value.**  Let $\mathcal{P}$ be the set of parameters, and let $f(S)$ be a performance function with $S \in 2^{\mathcal{P}}$. Assume that the following Non-negative Contribution and Replaceability conditions hold for parameter $k \in \mathcal{P}$:

1. **Non-negative Contribution:** The marginal contribution after adding a new parameter $k \notin S$ does not decrease. For $f$ representing the optimized performance in this paper, this condition is automatically satisfied.

$$\Delta(k|S) \equiv f(S \cup \{k\}) - f(S) \geq 0.$$

2. **Replaceability:** The marginal contribution of $k$ is zero in the presence of any parameter $l \notin S$:

$$f(S \cup \{l\} \cup \{k\}) - f(S \cup \{l\}) = 0.$$

Then the following hold

$$\Delta(l|S \cup \{k\}) \leq \Delta(l|S), \tag{2}$$

for all $l \notin S \cup \{k\}$ with $S \subset \mathcal{P} \setminus \{k\}$. In other words, pruning of $k$ increases the marginal contribution of other parameters.

*Proof.*

$$\begin{aligned}
\Delta(l|S \cup \{k\}) &= f(\{l\} \cup S \cup \{k\}) - f(S \cup \{k\}) \\
&= f(\{l\} \cup S \cup \{k\}) - f(S \cup \{k\}) + \{f(S) - f(S)\} \\
&\leq f(\{l\} \cup S \cup \{k\}) - f(S) \quad (\because f(S \cup \{k\}) - f(S) \geq 0, Assumption 1) \\
&= f(\{l\} \cup S \cup \{k\}) - f(S) + \{f(\{l\} \cup S) - f(\{l\} \cup S)\} \\
&= f(\{l\} \cup S) - f(S) \quad (\because f(\{l\} \cup S \cup \{k\} - f(\{l\} \cup S) = 0, Assumption 2) \\
&= \Delta(l|S).
\end{aligned}$$

$\square$

**Performance Retention under Pruning.**  With subsets $T \subseteq \mathcal{P} \setminus \{l\}$ and $S \subseteq \mathcal{P} \setminus \{l, k\}$, the SV of $l \in \mathcal{P}$ can be written as

$$\phi_l = \sum_{T \subseteq \mathcal{P} \setminus \{l\}} \underbrace{\frac{|T|!\,(n - |T| - 1)!}{n!}}_{w_T} \Delta(l \mid T) \tag{3}$$

$$= \sum_{S \subseteq \mathcal{P} \setminus \{l,k\}} \left[ \underbrace{\frac{|S|!\,(n - |S| - 1)!}{n!}}_{w_S} \Delta(l \mid S) + \underbrace{\frac{|S + 1|!\,(n - |S| - 2)!}{n!}}_{w_{S \cup \{k\}}} \Delta(l \mid S \cup \{k\}) \right]. \tag{4}$$

Let's define

$$w_S' \equiv w_S + w_{S \cup \{k\}} = \frac{|S|!\,(n - |S| - 2)!}{(n - 1)!}. \tag{5}$$

Then the SV for $l$ is always less than the SV for $l$ after pruning $k$.

$$\phi_l = \sum_{S \subseteq \mathcal{P} \setminus \{l,k\}} \left[ w_S\, \Delta(l \mid S) + w_{S \cup \{k\}}\, \Delta(l \mid S \cup \{k\}) \right] \tag{6}$$

$$\leq \sum_{S \subseteq \mathcal{P} \setminus \{l,k\}} \left( w_S + w_{S \cup \{k\}} \right) \Delta(l \mid S) \quad (\text{Eq. 2}) \tag{7}$$

$$= \sum_{S \subseteq \mathcal{P} \setminus \{l,k\}} w_S'\, \Delta(l \mid S) \tag{8}$$

$$= \phi_l^{\widetilde{\;}k}, \tag{9}$$

where

$$\phi_l^{\sim k} = \sum_{S \subseteq \mathcal{P} \backslash \{l,k\}} \underbrace{\frac{|S|!\,(n-|S|-2)!}{(n-1)!}}_{w_S'} \Delta(l \mid S).$$

Summing this inequality over all remaining parameters yields

$$J_{\text{tot}}(\mathcal{P} \backslash \{k\}) = \sum_{l \in \mathcal{P} \backslash \{k\}} \phi_l^{\sim k} \geq \sum_{l \in \mathcal{P} \backslash \{k\}} \phi_l = J_{\text{tot}}(\mathcal{P}) - \phi_k. \tag{10}$$

Therefore, under the Non-negativity contribution and replacability conditions, performance degradation by removing parameter $k$ is compensated by other remaining parameters and always less than $k$'s SV.

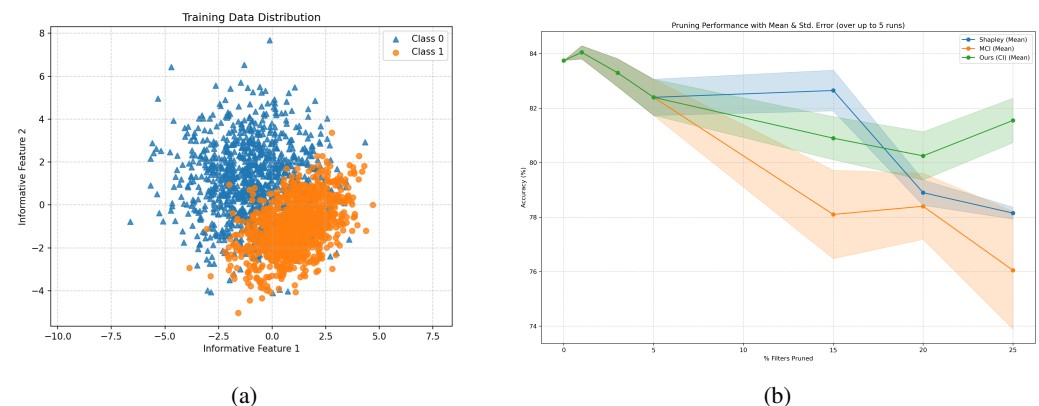

(a)                                                        (b)

Figure 2: (a) Visualization of 100-dimensional training data on the 2-D plane. (b) Pruning results for SV, MCI and CI.

## C   APPENDIX C

**Synthetic Experiment.**   We use two 100-dimensional Gaussians, where the first two dimensions are shown in Figure 2a and the remaining 98 dimensions are filled with standard Gaussian noise. We use a fully connected neural network with three layers, each containing 16 hidden units. The data is designed to be easily classified so that the pruned neural network possesses only enough capacity to identify simple decision boundaries. The two-level approximation scheme is used. We sample 1,000 permutations and 100 vertices for a regression function. For each vertex sample, the corresponding model is trained for 10 epochs with learning rate (0.01). For $f^*$, the optimized cross entropy loss is used.

**Results.**   The pruning results are presented in Figure 2b. Here, we report the mean and standard error of test accuracy from 5 independent simulations. The pruning ratio is defined as the ratio of the number of pruned parameters to the total number of parameters. While all three methods perform reasonably well at low pruning ratios (¡5%), the MCI methods extremely fail to capture the discriminative structure after 5%. Thus, the MCI method is excluded from baselines of real-world experiments.

## D APPENDIX D

Proof of Convergence for Cooperation Index

Let $CI_{true} \in [0, 1]$ be the true Cooperation Index for the parameter $i$. We estimate the CI value using Monte Carlo sampling with $N$ permutations.

Let $X_k$ be the random variable for the parameter $i$ and $k$-th sampled permutation. defined as:

$$X_k = \mathbf{1}(\Delta_{\pi_k,i} > \hat{\phi}_i). \tag{11}$$

where $\Delta_{\pi_i}$ is the marginal contribution in permutation $\pi_j$ and $\phi_i$ is the SHAP value of parameter $i$. Since $X_k$ takes values in $\{0, 1\}$, it follows a Bernoulli distribution. with probability of cooperation $p = CI_{true}$:

$$X_k \sim \text{Bernoulli}(p), \quad where \ \ p = CI_{true}. \tag{12}$$

The expectation and variance of a single sample $X_k$ are:

$$\mathbb{E}[X_k] = p = CI_{true}, \quad \text{Var}(X_k) = p(1-p) = CI_{true}(1 - CI_{true}). \tag{13}$$

### D.1 PROPERTIES OF THE ESTIMATOR

We define the estimator $\widehat{CI}$ as the mean of $N$ permutation samples mean of $N$ i.i.d samples:

$$\widehat{CI} = \frac{1}{N} \sum_{k=1}^{N} X_k = \overline{X}. \tag{14}$$

We set the estimator $\widehat{CI}$ as a new random variable $\overline{X}$ with the following properties:

- **Expectation:**

$$\mathbb{E}[\overline{X}] = \mathbb{E}\left[\frac{1}{N} \sum_{k=1}^{N} X_k\right] = \frac{1}{N} \sum_{k=1}^{N} \mathbb{E}[X_k] = p = CI. \tag{15}$$

- **Variance:**

$$\text{Var}(\overline{X}) = \text{Var}\left(\frac{1}{N} \sum_{k=1}^{N} X_k\right) = \frac{1}{N^2} \sum_{k=1}^{N} \text{Var}(X_k) = \frac{N \cdot p(1-p)}{N^2} = \frac{N \cdot CI(1 - CI)}{N^2}. \tag{16}$$

### D.2 APPLICATION OF CHEBYSHEV'S INEQUALITY

Using the Chebyshev's inequality for any random variable $X$ with finite expected value $\mu$ and non-zero variance $\sigma^2$, and for any real number $\epsilon > 0$, we can write the follows inequality:

$$P(|X - \mu| \geq \epsilon) \leq \frac{\text{Var}(X)}{\epsilon^2}. \tag{17}$$

Substituting $X = \overline{X} = \widehat{CI}$, $\mu = \mathbb{E}[\overline{X}] = CI$, and $\text{Var}(Y) = \text{Var}(\overline{X}) = \frac{CI(1-CI)}{N}$ into the inequality:

$$P(|\widehat{CI} - CI_{true}| \geq \epsilon) \leq \frac{CI_{true}(1 - CI_{true})}{N\epsilon^2}. \tag{18}$$

### D.3 BOUNDING THE VARIANCE

The term $CI(1 - CI)$ is a quadratic function of $CI$, which achieves its maximum value at $CI = 0.5$. Thus, the variance is upper bounded by:

$$CI(1 - CI) \leq 0.25 \quad \text{for all } CI \in [0, 1] \tag{19}$$

Substituting this upper bound into the inequality derived in Eq(8), we obtain the bound of the estimator $\widehat{CI}$:

$$P(|\widehat{CI} - CI| \geq \epsilon) \leq \frac{1}{4N\epsilon^2} \tag{20}$$

This result demonstrates that the upper bound depends only on the error margin $\epsilon$ and number of sample $N$, and is independent of the model size (parameter dimension-free). Therefore, the Cooperation Index converges even if the model scales up.

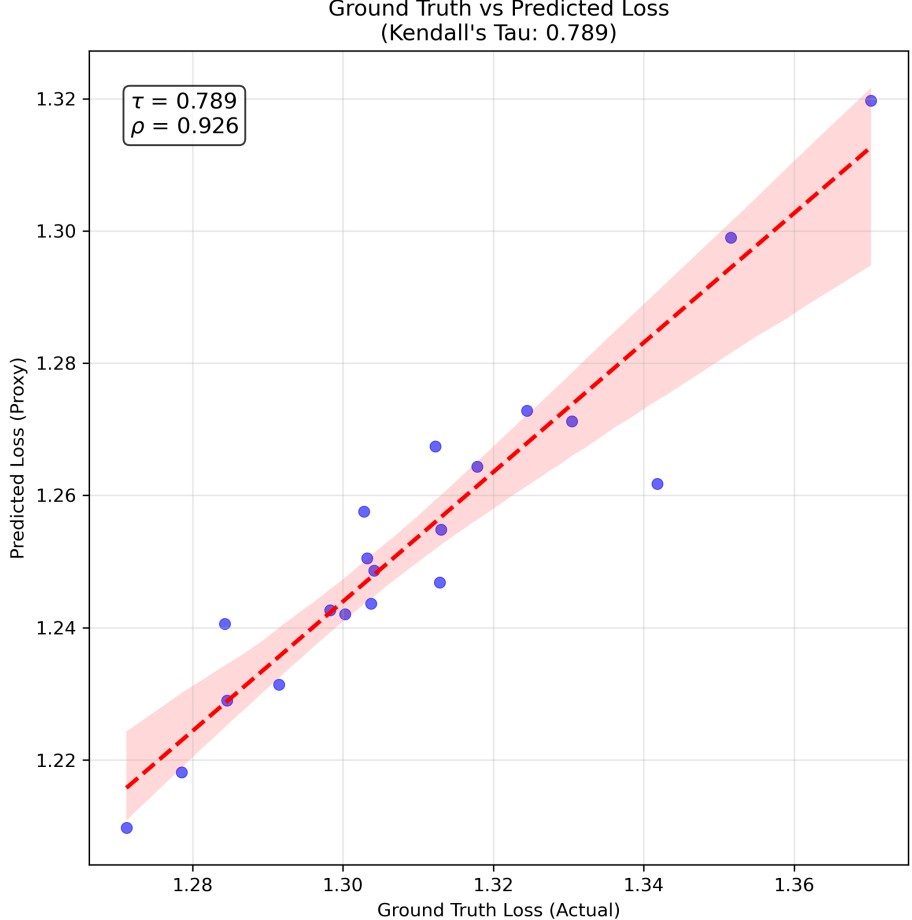

## E    APPENDIX E

Validation of the regression function for approximating vertex values via rRank correlation constant kendall's Tau and Spearman's rho. The regression function for VGG-16 model reasonably preserves the relative importance ranking required for CI.