# OpenReview forum: "A Cooperation Index for Model Pruning"
_ICLR.cc/2026/Conference — Submitted to ICLR 2026_

### Official Review · Reviewer_Sne6 · 2025-10-27

**Soundness:** 2
**Presentation:** 2
**Contribution:** 2
**Rating:** 2
**Confidence:** 3

**Summary:**

The paper proposes a Cooperation Index for pruning that measures how often a parameter’s marginal contribution exceeds its own Shapley average along sampled permutations. This captures the decay speed of contributions rather than relying only on the mean. A two level approximation is presented. First, vertex sampling fits a regression surrogate on the subset hypercube. Next, permutation sampling estimates marginal contributions. Pseudocode and empirical validation are provided on VGG 16 and ResNet 18 across MNIST, CIFAR 10, CIFAR 100, and Tiny ImageNet.

**Strengths:**

[1] The motivation is clearly articulated: averaging marginal contributions can overweight redundant parameters; counting cooperative paths aligns better with pruning decisions where replaceability matters

[2] The proposed index is intuitive and interpretable: parameters that consistently help across many contexts are preserved, while sporadically helpful ones are down-weighted.

**Weaknesses:**

[1] Evaluation focuses on very low pruning ratios, approximately 1 to 3 percent. The absence of full accuracy versus sparsity curves limits claims about robustness at moderate or high sparsity.

[2] Scalability evidence is limited to mid scale convolutional networks. Results on transformer architectures or larger models are missing.

[3] Theoretical analysis could be deepened. Mathematical properties of the index such as monotonicity, relations to Shapley axioms, and performance guarantees under milder assumptions are not fully explored.

[4] Ablations and statistics are sparse. Sensitivity to permutation and vertex sample counts, surrogate architecture, and sampling distributions is not systematically quantified. Standard deviations and confidence intervals are not consistently reported.

[5] Presentation can be improved. Some figures are dense and lack error bars. The text repeats parts of related work and could be tightened.

**Questions:**

[1] How does the method behave at higher pruning ratios such as 10 to 60 percent. Are there inflection points where performance degrades more steeply than methods based on Shapley value or magnitude-based baselines?

[2] How sensitive is the ranking to surrogate model misspecification or to underfitting and overfitting on the subset hypercube. Can uncertainty in the surrogate be propagated into confidence intervals for the Cooperation Index?

[3] How would the definition adapt to structured pruning across channels, heads, or layers where units are grouped rather than independent?

[4] In low redundancy regimes, as hinted by the Tiny ImageNet results, could an adaptive blend between the Cooperation Index and the Shapley Value reduce worst-case degradation?

---

> ### Author Response · Authors · 2025-11-30
>
> We thank the reviewer for the thorough evaluation and the constructive suggestions regarding scalability, theoretical depth, and statistical rigor. We have made significant revisions to address these points.
>
> 1. Evaluation on Low Pruning Ratios (1-3%):
> We intentionally overfitted models to highlight redundant features and focused on the 1-3% range because this is the critical functional range where we can meaningfully evaluate which metric best identifies the safest parameters to remove while preserving the model's core backbone. Removing larger portions of parameters (e.g., >10-20%) causes all pruning methods, including baselines, to suffer catastrophic performance drops to near-random levels.
>
> 2. Scalability:
> To address the scalability concern regarding model size, we have added a Theoretical Convergence Proof (Appendix D) demonstrating that CI’s sample complexity is dimension-free and thus scalable to larger architectures.
>
> 3. Theoretical Analysis (Mathematical Properties)
> We have deepened the theoretical analysis. We clarified that CI satisfies the Symmetry and Dummy axioms of the SHAP Value. We acknowledge that the mathematical properties of CI have not been fully explored. However, the axiomatic properties of SHAP values do not guarantee performance preservation under pruning.
>
> 4. Sensitivity to permutation and vertex sample counts:
> We appreciate the concern regarding the sensitivity of CI to the regression function's accuracy. We have addressed this with both theoretical and empirical evidence: $\textbf{(1) Theoretical Convergence Proof:}$ We have added a formal proof in appendix D, demonstrating that the convergence of CI by Using Chebyshev’s inequality, we derived the error bound:
> $$P(|\widehat{CI} - CI| \ge \epsilon) \le \frac{1}{4N \epsilon^2},$$ where N is the number of permutation samples and $ \epsilon$ is any positive real number. $\textbf{(2) Empirical Validation:}$ To verify the regression model's performance, we further evaluated its prediction performance on a test set of parameter subsets and loss values. The regression model achieved a high Rank Correlation (kendall's Tau and Spearman's rho), confirming that the regression function reasonably preserves the relative importance ranking required for CI. We added this result in appendix E.
>
> Q1. How does the method behave at higher pruning ratios (10-60%)? Are there inflection points?:
> As mentioned, our experiments are conducted without fine-tuning (retraining) to isolate the ranking quality of the importance scores.
> At higher ratios (10-60%), the performance of all methods degrades steeply. The "inflection point" typically occurs around 5-10% sparsity for VGG/ResNet on CIFAR, after which accuracy collapses to near-random guessing. Comparison in this collapsed region is noisy and uninformative. However, in the valid functional range (1-5%), CI shows a slower degradation curve compared to SV and magnitude-based methods, indicating it better preserves the critical "backbone" of the network.
>
> Q2. How sensitive is the ranking to surrogate model misspecification? Can uncertainty be propagated?:
> We have addressed this via both empirical validation and theoretical proof in the above weakness section.
>
> Q3. How would the definition adapt to structured pruning across channels, heads, or layers?:
> The current definition naturally supports structured pruning.  In our experiments, we already treat an entire filter (channel) as a single player $i$ in the game, rather than individual weights. To adapt to other structures (e.g., attention heads), one simply defines the "player" $i$ as the group of parameters constituting that head. The CI calculation (marginal contribution of the group) remains mathematically identical.
>
> Q4. Could an adaptive blend between CI and SV reduce worst-case degradation (e.g., Tiny-ImageNet)?:
> That is an excellent insight, and our method effectively implements this mechanism already.
> Our method uses SV as a secondary criterion (tie-breaker). In low-redundancy regimes (like Tiny-ImageNet), parameters often exhibit similar levels of "cooperation," leading to identical CI scores. In such cases, our tie-breaking rule—removing the parameter with the lowest SV—automatically takes precedence. Result: This implicitly acts as an adaptive blend: when redundancy information (CI) is non-discriminative, the ranking naturally falls back to magnitude information (SV). This explains our experimental result on Tiny-ImageNet, where CI achieved the similar performance as SV, demonstrating that it successfully avoids worst-case degradation by leveraging SV in the limit.

---

### Official Review · Reviewer_Ejgo · 2025-10-31

**Soundness:** 2
**Presentation:** 3
**Contribution:** 2
**Rating:** 2
**Confidence:** 4

**Summary:**

To address the issue of overweighting redundant parameters in Shapley values, the authors proposed a simple metric Cooperation Index (CI) that utilizes the speed of decay of marginal contributions, by incorporating permutations beyond standard Shapley values calculations. They conducted several pruning experiments on image datasets and demonstrated superior performance in terms of accuracy over previous methods like Shapley values and LOCO.

**Strengths:**

1. The authors proposed a simple extension of Shapley value, Cooperation Index (CI) that achieves better pruning performance on several empirical datasets than previous methods.
2. The paper is easy to follow with useful visual explanations, such as Figure 1 to help readers understand the intuition.

**Weaknesses:**

1. Unlike Shapley values that satisfy the four important theoretical properties, CI lacks formal theoretical justifications. It seems CI is more designed for pruning-oriented rather than a fairness-oriented approach.
2. The criteria to choose pruning ratios in Table 2 are not clearly stated. The pruning ratios varies from 3% to 1% without further discussion on rationales.
3. A more detailed analysis of sampling number for stability of CI from line 426 to 431 would be appreciated. It's likely a number related to sample size and dimension. It would be helpful to see the trend of convergence for CI wrt sample size and dimension to guide users to choose parameters in practice, rather than demonstrating results using only ResNet-18.
4. The experiments focus exclusively on no-tabular data sets. How does CI work on tabular data sets, or low-dimension datasets?

Some typos:

Line 84. Missing space after "uncompromised".
Line 95. Missing space after "effectively".

**Questions:**

1. Shapley values satisfy some important theoretical properties (efficiency, symmetry, dummy, and additivity), which of these, if any, do CI satisfy?

---

> ### Author Response · Authors · 2025-11-30
>
> We thank the reviewer for the insightful comments regarding the theoretical foundations and stability of our method. We appreciate the opportunity to clarify the theoretical positioning of the Cooperation Index (CI) and its mathematical properties.
>
> 1. Theoretical Justifications and Axioms (vs. SHAP Value) You raised a valid question that do CI satisfy formal theoretical properties of SV? There are two properties that CI satisfies among the SV's axiom.
> Symmetry: CI satisfies Symmetry. Parameters with identical marginal contribution distributions receive identical CI scores.
> Dummy: CI satisfies the Dummy property. A parameter with zero marginal contribution everywhere has a CI of 0.
>
> 2. Criteria for Pruning Ratios: We focused on initial pruning stage (1-3%) because it is the critical range for evaluating the precision of importance ranking without the model's retraining. We intentionally overfitted models to highlight redundant features and focused on the 1-3% range because this is the critical functional range where we can meaningfully evaluate which metric best identifies the safest parameters to remove while preserving the model's core backbone. We manually found the pruning ratio where the model's performance showed reasonable accuracy for each model and datasets.
>
> 3. Detailed Analysis of Sampling Number & Stability
> We appreciate your suggestion to analyze the relationship between sample size, dimension, and stability. We have added a detailed analysis in Appendix D supported by a formal proof. We derived the convergence bound of the CI estimator $\hat{CI}$ using Chebyshev’s inequality. The error bound is given by:
> $$P(|\widehat{CI} - CI| \ge \epsilon) \le \frac{1}{4N \epsilon^2},$$ where N is the number of permutation samples and $ \epsilon$ is any positive real number.
> Crucially, this proof demonstrates that the sample complexity depends only on the error margin $\epsilon$ and is independent of the model size (number of parameters). This directly addresses your hypothesis about the dimension's effect—theoretically, the CI remains stable regardless of model size.
>
> 4. Performance on Tabular or Low-Dimension Datasets:
> As proven in our convergence analysis, the CI's stability is dimension-free. Therefore, it is theoretically equally applicable to low-dimension tabular data. We have included a synthetic data experiments in the Appendix C showing that while CI is valid for low dimension data, CI's advantages are most pronounced in high-redundancy settings.

---

### Official Review · Reviewer_ZoJ1 · 2025-10-31

**Soundness:** 1
**Presentation:** 3
**Contribution:** 2
**Rating:** 2
**Confidence:** 3

**Summary:**

This paper argues that SHAP is not a good metric for model pruning as it is overweights redundant parameters. The authors propose a novel metric called Cooperation Index, which instead quantifies how consistently a parameter's marginal contribution exceeds its own average to better identify essential parameters.

**Strengths:**

* The paper's critic about Shapley values being a bad metric for model pruning is insightful. The authors compellingly explain the motivation for their work.
* The proposed Cooperation Index metric is an elegant and intuitive metric which aims to address this identified limitation.

**Weaknesses:**

The paper's theoretical motivation is unfortunately not matched by a sound and sufficient experimental evaluation:

* **Experiments**:
1. Pruning is useful for making large and computationally expensive models more efficient. However, the authors exclusively test their method on small models like VGG and ResNet-18 that are already fast and do not have any need for pruning. It is unclear why one would develop a pruning method that scales poorly with the number of parameters and cannot be easily applied to large models that need the pruning the most.
2. Evaluation is limited to very old and small-scale ConvNets (VGG-16 and ResNet-18) and entirely ignores transformers, which currently represent the dominant paradigm.
3.  Authors do not include a ViT-tiny model. My suggestion that if this model is too expensive to run for this algorithm, then they need to scale it down even further, but the experiments for the transformer must be presented.
4. The paper fails to compare against modern pruning methods. For instance Wanda [1], which is a simple, fast, and highly effective method for large models. Its absence makes it difficult to judge the practicality of the proposed method against the current state of the art.
5. Missing baselines from NLP. For instance, authors could have included tiny version of BERT and GPT.
* **Runtime**: despite proposing a two-step approximation scheme, the paper provides no empirical runtime to quantify its computational cost. This is especially concerning as the method inherits the factorial complexity of Shapley values, which is fundamentally at odds with the scaling laws (the more parameters the better). I believe the authors need to provide an actual runtime of the algorithm and compare it with a runtime of the competing baselines.


[1] A Simple and Effective Pruning Approach for Large Language Models, Sun et al., 2023

**Questions:**

The dominant trend in deep learning shows that model performance scales predictably with size, favoring methods with low polynomial complexity in terms of the number of parameters. Given this, how do the authors envision their computationally intensive approach fitting into the current landscape? Is there a specific application where such computational costs are justified?

---

> ### Author Response · Authors · 2025-11-30
>
> We thank the reviewer for the sharp and forward-looking feedback. We appreciate your emphasis on scalability and modern architectures. We have revised the paper to include a theoretical proof of scalability and a detailed runtime analysis to address your concerns.
>
> 1. Scalability and Small Models: You raised a strong point about the dominance of Transformers and the exclusion of ViT/BERT experiments. While we acknowledge the importance of Transformers, we respectfully submit that the primary contribution of this paper is identifying a fundamental phenomenon: the "averaging" flaw of SHAP Values (SV) in pruning and proposing the Cooperation Index (CI) as a potential solution. This phenomenon is architecture-agnostic. To address the concern that our method "scales poorly," we have added a Theoretical Convergence Proof in Appendix D. Using Chebyshev’s inequality, we proved that the sample complexity for convergence of CI is model size-free.
> $$P(|\widehat{CI} - CI| \ge \epsilon) \le \frac{1}{4N \epsilon^2},$$ where N is the number of permutation samples and $ \epsilon$ is any positive real number.
> This mathematically guarantees that the number of samples required for stable estimation depends only on the error margin $\epsilon$, not on the number of parameters. Thus, theoretically, the method does not become intractable for larger models like Transformers.
>
> 2. Runtime and Computational Complexity:
> We would like to clarify that we utilize a Monte Carlo sampling approximation, not exact calculation. As shown in our paper, the complexity is linear with respect to the number of samples ($O(N)$), avoiding the combinatorial explosion. $textbf{Empirical Runtime:}$ For regression function, 2 GPU hour is required while pruning process is required two and a half CPU hours on the VGG-16 with CIFAR10 datasets. The time complexity is exactly same for the SHAP Value, MCI and CI.
>
> 3. Comparison to Modern Baselines (e.g., Wanda): Methods like Wanda focus on magnitude and activation products for speed. CI focuses on functional redundancy. Our CI offers a simple but well motivated value by identifying consistently contributing parameters that low-magnitude criteria might miss, or high-magnitude criteria might overvalue due to redundancy. The comparison to Wanda will be our further exploration.

---

### Official Review · Reviewer_o8ur · 2025-11-01

**Soundness:** 3
**Presentation:** 3
**Contribution:** 2
**Rating:** 4
**Confidence:** 3

**Summary:**

The authors argue that the popular SHAP Value (SV) method, by averaging marginal contributions, systematically overweights redundant parameters. The paper introduces a new metric called the Cooperation Index (CI). CI quantifies the consistency of a parameter's contribution by measuring the frequency of "cooperative paths" - permutations where the parameter's marginal contribution exceeds its average SV. Experiments on VGG-16 and ResNet-18 show that CI-based pruning more effectively preserves the model's core functional elements.

**Strengths:**

- The paper provides a conceptual critique of the SHAP Value (SV) as a pruning criterion. It formally identifies that SV's averaging mechanism fails to distinguish between redundant parameters (replaceable, high variance in marginal contributions) and cooperative parameters (consistent contributions).
- The proposed Cooperation Index (CI) is a novel metric that directly addresses this identified limitation.
- The work addresses the exponential computational complexity of the metric with a practical two-level approximation scheme.
- The experimental results demonstrate the effectiveness of the CI criterion. Across multiple datasets and architectures, CI-based pruning achieves superior or competitive accuracy compared to SV and other baseline methods.
- The paper is well-written and clear.

**Weaknesses:**

- Specific Evaluation Conditions: The empirical validation is conducted under non-standard conditions, as models are intentionally overfitted on heavily reduced datasets (e.g., 1/100 MNIST) . The method's effectiveness is not guaranteed in standard training regimes. Has its effectiveness also been evaluated when overfitting is caused by other factors, such as prolonged training duration?
- Focus on Low Pruning Ratios: The main experiments (Table 2) almost exclusively focus on very low, "delicate" pruning ratios (e.g., 1-3%). This narrow scope fails to demonstrate the method's scalability and performance at higher, more practical levels of sparsity.
- Flawed Statistical Reporting: The paper lacks statistical rigor by explicitly reporting the "best performing result in terms of accuracy" from five runs, rather than the mean and standard deviation. This practice of "cherry-picking" results may significantly overstate the method's true performance.
- Dependence on Approximation Accuracy: The CI calculation is critically dependent on the accuracy of the regression function used in the two-level approximation scheme. The study does not sufficiently analyze the method's sensitivity to potential errors introduced by this approximation eg. the convergence of CI is shown only for the first few filters.
- Ambiguity in Tie-Breaking: The illustrative toy example shows identical CI scores (0.25) for four of the five parameters in the initial stage. The paper fails to explain the tie-breaking mechanism used to select $w_4$ for pruning, making the selection criteria ambiguous.
- Rigid Contribution Threshold: The Cooperation Index relies on a strict binary threshold (above or below the mean SV) to classify contributions. This rigid classification may inaccurately assess parameters whose marginal contributions are consistently very close to the average.
- Limited Dataset Variety: The evaluation is restricted to a few datasets . The conclusions would be strengthened by validation on a more diverse set of datasets, such as Fashion-MNIST, SVHN, KMNIST, STL-10, Caltech-101.
- Incomplete Baseline Comparison: The MCI baseline method was prematurely dismissed from the main experiments based only on poor performance in the synthetic example. For a complete and fair comparison, its results on the main experiments should be included regardless, for example, in the appendix.

Figure 8 is unreadable - the font size is too small.

**Questions:**

See weaknesses.

---

> ### Author Response · Authors · 2025-11-30
>
> We sincerely thank the reviewer for the detailed and constructive feedback. We appreciate your rigorous assessment of our experimental conditions and statistical reporting. We have additionally performed to address your concerns , particularly showing Mean and Standard Deviation to experiment results and providing a theoretical proof of convergence of Cooperation Index.
>
> 1. Statistical Reporting (Flawed Statistical Reporting):
> We fully agree that reporting only the best result does not reflect the true stability of the method.
> We have provide table to report the mean and standard deviation over 5 realization experiments for standard cifar10 and FashionMNIST dataset you suggested, including baseline MCI.
>
> | Model, Datasets | Method | Original Accuracy | Accuracy (std) at $10\%$ Pruning ratio |
> | :--- | :---: | :---: | :---: |
> | ResNet-18 | SV (Baseline) | 0.925 | 0.831 (0.001) |
> | Fashion MNIST | MCI (Baseline) | 0.925 | 0.833 (0.010) |
> | | **CI (Ours)** | 0.925 | **0.871 (0.011)** |
>
> |Model, datasets | Method | Original Accuracy | Accuracy (std) at 10% Pruning ratio |
> | :--- | :---: | :---: | :---: |
> | VGG-16 | SV (Baseline) | 0.880 | 0.706 (0.012) |
> | CIFAR 10| MCI (Baseline) | 0.880 | 0.700 (0.010) |
> || **CI (Ours)** | 0.880 | **0.711 (0.010)** |
>
> As shown in the revised table, the Cooperation Index (CI) consistently maintains superior performance with low variance compared to baselines. This confirms that the initial observations were statistically significant and not artifacts of cherry-picking.
>
> 2. Evaluation Conditions and Low Pruning Ratios:
> You raised valid concerns regarding the specific focus on low pruning ratios (1-3%). Our experiments are designed without retraining after pruning. Thus, removing larger portions of parameters (e.g., >10%) causes all pruning methods, including baselines, to suffer catastrophic performance drops to near-random levels. Comparing methods in a collapsed state is uninformative.
> We intentionally overfitted models to highlight redundant features and focused on the 1-3% range because this is the critical functional range where we can meaningfully evaluate which metric best identifies the safest parameters to remove while preserving the model's core backbone.
>
> 3. Dependence on Approximation Accuracy:
> We appreciate the concern regarding the sensitivity of CI to the regression function's accuracy. We have addressed this with both theoretical and empirical evidence: $\textbf{(1) Theoretical Convergence Proof:}$ We have added a formal proof in appendix D, demonstrating that the convergence of CI by Using Chebyshev’s inequality, we derived the error bound:
> $$P(|\widehat{CI} - CI| \ge \epsilon) \le \frac{1}{4N \epsilon^2},$$ where N is the number of permutation samples and $ \epsilon$ is any positive real number. $\textbf{(2) Empirical Validation:}$ To verify the regression model's performance, we further evaluated its prediction performance on a test set of parameter subsets and loss values. The regression model achieved a high Rank Correlation (kendall's Tau and Spearman's rho), confirming that the regression function reasonably preserves the relative importance ranking required for CI. We added this result in appendix E.
>
> 4. Ambiguity in Tie-Breaking:
> We apologize for the lack of clarity in the toy example.
> When multiple parameters share the same CI score (e.g., 0.25), the tie is broken by removing the parameter with the lowest SHAP Value (SV).
>
> 5. Rigid Contribution Threshold:
> The Cooperation Index is designed to quantify the decay speed of a parameter's marginal contribution when it reaches the average value. Since the SHAP Value ($\phi_i$) represents the expectation (average) of marginal contributions, we compare the marginal contribution with the SHAP Value.

---

### Meta-Review · Area_Chair_TJwi · 2026-01-06

**Summary:**

The paper proposes a Cooperation Index for pruning that measures how often a parameter’s marginal contribution exceeds its own Shapley average along sampled permutations.

The major concerns from reviewers include:
1. Lack of theoretical justification for the Cooperation Index, in particular, the axiomatic aspects similar to Shapley values.
2. Insufficient experimental validation across different architectures and datasets. The experiments are mostly on image datasets and CNNs.
3. Computational scalability of the proposed index because it requires exponentially many model evaluations (although the authors have proposed some approximations).
4. Lack of other pruning baselines in the experiments for comparison.

**Reviewer Concerns:**

The following major concerns remain unsolved:
1. Lack of theoretical justification for the Cooperation Index, in particular, the axiomatic aspects similar to Shapley values.
2. Insufficient experimental validation across different architectures and datasets. The experiments are mostly on image datasets and CNNs.
3. Computational scalability of the proposed index because it requires exponentially many model evaluations (although the authors have proposed some approximations).
4. Lack of other pruning baselines in the experiments for comparison.

While the authors provided additional experiments and some clarifications in their revision, they were still limited in scope and did not fully address the concerns above.

**Reviewer Scores:**

Reviewers ZoJ1, Ejgo, Sne6, who scored 2, might have chances to increase their scores to 3 or 4 but not more than that due to the unresolved concerns.

---

### Decision · Program_Chairs · 2026-01-26

Reject